# Comparing standard and technology-assisted peer-delivered CBT for perinatal depression: A causal mediation study

Ahmed Waqas[1] ●, Nadine Seward[2], Najia Atif[3] ●, Abid Malik[4] ●, Anum Nisar[1], Siham Sikander[1] ●, Huma Nazir[3], Duolao Wang[5] and Atif Rahman[1] ●

[1]Department of Primary Care & Mental Health, Institute of Population Health, University of Liverpool, Liverpool, UK; [2]Division of Psychiatry, Centre for Clinical Brain Sciences, Univeristy of Edinburgh, Edinburgh, UK; [3]Human Development Research Foundation, Islamabad, Pakistan; [4]Department of Public Mental Health, Health Services Academy, Islamabad, Pakistan and [5]Global Health Trials Unit, Liverpool School of Tropical Medicine, Liverpool, UK

## Research Article

**Keywords:**
perinatal depression; social support; mediation; cognitive-behavioural therapy; CBT; digital CBT

**Corresponding author:**
Ahmed Waqas;
Email: ahmed.waqas@liverpool.ac.uk

## Abstract

The ENHANCE non-inferiority trial that took place in a deprived setting in Pakistan demonstrated that a technology-assisted digital adaptation of the Technology Assisted Thinking Healthy Programme (THP-TAP) was no different than the face-to-face THP in improving symptoms of perinatal depression. The present study examines the mechanisms through which THP-TAP improved symptoms of perinatal depression (or not) compared to the face-to-face THP.

We applied a counterfactual-based approach to mediation – particularly interventional effects – to decompose the total effect of the THP-TAP intervention on symptoms of perinatal depression into the following pre-specified indirect effects: number of sessions attended; behavioural activation; perceived social support; problem-solving and cognitive-restructuring skills; and peer empathy. Mediators were assessed at 3 months post-partum, and depressive symptoms were measured at 6 months using the Patient Health Questionnaire-9 (PHQ-9).

Perceived social support in THP-TAP arm mediated an improvement in symptoms of perinatal depression compared to the standard face-to-face THP group (adjusted mean difference in PHQ-9 scores attributable to perceived social support in the technology-assisted digital adaptation of the THP group compared to the World Health Organisation THP group: −0.072, bias-corrected 95% confidence interval: −0.170, −0.018). There was no difference to support the indirect effects for all other mediators.

Even in the absence of treatment superiority, our findings suggest that levels of perceived social support were an important feature of the THP-TAP intervention, which resulted in improved symptoms of perinatal depression. From a practical perspective, these findings highlight the importance of social connectedness as a mechanism of change, demonstrating that peer-delivered digital psychosocial interventions can successfully cultivate this relational component.

## Impact statement

A co-designed, app-supported version of the World Health Organization Thinking Healthy Programme (THP-TAP) achieved perinatal depression outcomes comparable to standard delivery, supporting scalable and equitable access. Mediation analysis indicated that perceived social support, from significant others, family and friends, was the sole validated pathway to symptom improvement, with a larger indirect effect in the digital THP-TAP than in face-to-face delivery. Neither greater session attendance nor additional cognitive-behavioural therapy skill content conferred extra benefits.

Therefore, digital enhancements should prioritise mobilising and sustaining real-world support networks (family engagement, peer connectors and community prompts) rather than increasing platform "dose" or expanding curricula.

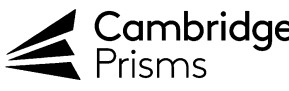



## Background

Perinatal depression is a global health priority because it compromises maternal well-being, infant neuro-development and long-term human capital, especially in low- and middle-income countries (LMICs) (Gelaye et al., 2016; Atif et al., 2021). Pakistan illustrates the magnitude of the challenge with community prevalence estimates of 25–37%, far exceeding regional averages, yet specialist mental-health resources are scarce and primary-care services are already overstretched

(Gelaye et al., 2016). Consequently, the task-sharing strategies employing lay health workers to deliver evidence-based psychosocial care have become central to maternal-health policy in LMICs.

The Thinking Healthy Programme (THP) is the most extensively tested task-shared intervention for perinatal depression in LMICs (Rahman et al., 2008). It integrates brief cognitive-behavioural therapy (CBT) techniques (behavioural activation, thought challenging and problem solving) with strategies to mobilise family support. Early cluster-randomised trials in Pakistan demonstrated large effect sizes when the standard World Health Organisation THP (WHO-THP) was delivered face-to-face by educated Lady Health Workers (LHWs), a government cadre with 15 months of formal training (Rahman et al., 2008). However, scaling up the LHW-delivered THP proved challenging, as LHWs are already responsible for family planning, vaccination and child health services (Hafeez et al., 2011). Attempts to scale THP through peer volunteers, however, revealed recurrent implementation hazards (Zafar et al., 2016; Atif et al., 2017; Rahman et al., 2019; Sikander et al., 2019; Rahman et al., 2021; Rahman et al., 2023).

To preserve fidelity while easing workforce pressure, our multi-disciplinary team co-produced with peers with lived experience, a digital adaptation of the WHO-THP in which animated, culturally tailored storylines present the CBT content on an Android application that functions offline (Atif et al., 2022). Crucially, the delivery agents are peers with lived maternal depression experience rather than LHWs. Automation serves three system-level purposes: (i) standardised multimedia modules insulate the active CBT ingredients from programme drift; (ii) interactive decision rules, pause prompts and competence checks provide "in-session" supervision, mitigating voltage drop when external oversight is scarce; and (iii) offloading psycho-education to the device shortens initial training and reduces the specialist supervisory burden.

From a health-systems perspective, this redesign realigned task-sharing with operational reality. Economic modelling indicated that, once tablets are amortised, mean delivery costs per treated woman fall below those of the LHW-only model because peer stipends are modest and ongoing specialist supervision is largely virtual (Rahman et al., 2023). Yet health-policy endorsement requires clear evidence that THP-TAP is at least as clinically effective as traditional WHO-THP delivered by educated LHWs. A single-blind, cluster-randomised non-inferiority trial in 70 rural villages addressed this question in a large non-inferiority cluster randomised controlled trial (cRCT) (Rahman et al., 2023) demonstrated that digital THP achieved 3-month post-partum remission rates of 92.3% in the technology-assisted peer arm versus 83.5% in the standard LHW arm. The digital THP simultaneously reduced supervisory demands, freeing scarce professional time for other public health priorities.

The primary policy question of effectiveness in real-world settings thus appears resolved, but the scientific question of how the two delivery models achieve their effects remains open. Competing theoretical perspectives attribute change to specific CBT skill acquisition to common therapeutic factors (empathy and hope) or to augmented social support (Huibers et al., 2021). Clarifying which mechanisms dominate in each modality is not merely academic. It has real-world implications for refining the digital THP – for instance, if automated CBT dosage is the principal mediator, future iterations should prioritise multimedia refinement; if peer warmth and network activation drive outcomes, investment should focus on recruiting and retaining high-empathy counsellors (Cuijpers, 2019;

Huibers et al., 2021; Waqas et al., 2023; Waqas et al., 2024; Liaquat et al., 2025; Seward et al., 2025).

Accordingly, our study aims to elucidate the mechanisms of change in THP-TAP by testing a causal mediation model that traces symptom reduction through both CBT-specific factors (acquisition and enactment of skills – behavioural activation, problem-solving and cognitive restructuring – measured via session-level adherence ratings and patient-reported activation) and common factors (peer–client empathic alliance and mobilisation of social support). Elucidating these pathways will move the field beyond "black-box" evaluations (Mulder et al., 2017), inform optimisation of both digital peer-facilitated and face-to-face LHW delivery platforms and generate actionable knowledge about the active ingredients of low-intensity CBT for perinatal depression in LMIC health system settings.

## Methods

### *Study design and rationale*

This mediation analysis builds upon a completed cluster RCT carried out in three subdistricts of Rawalpindi District, Punjab, Pakistan – Kallar Syeddan, Gujar Khan and Rawalpindi. It utilised a non-inferiority, stratified design comparing two delivery models of the WHO-THP: the standard THP delivered by LHWs and the THP-TAP delivered by trained community peers (Rahman et al., 2023). Given previous evidence confirming that LHW-delivered WHO-THP significantly reduces perinatal depression (Rahman et al., 2008), this study aimed to determine whether the peer-delivered technology-assisted THP was at least equally effective (Rahman et al., 2023). Details regarding the study design can be read in the protocol published elsewhere (Rahman et al., 2023).

### *Participant recruitment and eligibility*

Participants for this mediation analysis were women screened in the parent trial. Pregnant women in their second or third trimester identified through LHW registers were screened using the Urdu-translated Structured Clinical Interview for DSM-IV (SCID) to diagnose major depressive episodes (MDEs), a method extensively validated in previous studies in Pakistan (Rahman et al., 2008). Eligible participants were women aged 18 years or older, diagnosed with a current Diagnostic and Statistical Manual of Mental Disorders, 4th Edition (DSM-IV) MDE and planning to remain in the study area for at least 1 year. Women requiring inpatient psychiatric or medical care, unable to communicate in Urdu or having severe medical or psychiatric comorbidities necessitating specialised treatment were excluded. Participation was voluntary, and informed written consent was obtained after participants had a 24-h period to review study details.

### *Ethical considerations*

The authors assert that all procedures contributing to this work comply with the ethical standards of the relevant national and institutional committees on human experimentation and with the Helsinki Declaration of 1975, as revised in 2013. Ethical approval was secured from the Research Ethics Committees of the Human Development Research Foundation (Pakistan), the University of Liverpool (ethical approval: UoL001668) and the National Bioethics Committee (Pakistan). Individual-level informed consent and village-level permission from the District Health Office were

mandatory. Participant confidentiality and well-being were prioritised, with distress managed through empathetic communication and, when necessary, referrals to specialised mental health or social services. The University of Liverpool sponsored the trial and provided insurance coverage consistent with international clinical research standards. Confidentiality protocols ensured all personal information was securely stored, with no identifying details disclosed in any publications. Participants were informed that they could withdraw at any time without affecting their routine healthcare access.

### Masking and baseline assessments

While participants and intervention providers could not be blinded to allocation, outcome assessors remained masked to participants' treatment assignment. Participants were instructed not to disclose their assigned treatment arm during assessments, and any inadvertent disclosures were documented. Baseline assessments captured demographic data (age, education and socioeconomic status), obstetric history, depression severity, social support and domestic violence, conducted privately either in participants' homes or at designated LHW Health Houses.

### Intervention details

Interventions implemented in the parent trial included the standard WHO-THP, a manualized intervention based on CBT principles, delivered by LHWs through eight structured sessions from late pregnancy to 3 months postpartum (Rahman et al., 2008; Atif et al., 2022; Rahman et al., 2023). LHWs were trained over 5 days in communication skills, role-play exercises and the THP manual protocols. The THP-TAP was delivered by trained community peers via a multimedia Android application designed using Human-Centered Design methods. This app transformed WHO-THP content into culturally adapted animated modules. Peers, nominated from local communities, received a 3-day training covering communication techniques, CBT elements and operation of the tablet-based app. Animated modules included scripted scenarios and interactive "pause points," enabling peer facilitators to contextualise content for participants' everyday environments. Regular monthly supervision ensured intervention fidelity. Details regarding the development of the digital intervention are published elsewhere (Atif et al., 2022).

### Measurement of depression and mediators

Depressive symptoms were assessed using the Patient Health Questionnaire (PHQ-9), which has been validated in Pakistan with a cut-off score of ≥10 denoting probable depression (19). Administered at baseline, and 3 and 6 months post-partum, the PHQ-9 allowed us to capture continuous change in symptom severity and was preferred over a categorical SCID diagnosis because its greater variance enhances the statistical power of mediation analyses (Kroenke et al., 2001).

### Measures for mediators

Mediator selection was grounded in the cognitive-behavioural framework underpinning the WHO-THP, which posits that changes in cognition, behaviour and interpersonal support drive improvements in perinatal depressive symptoms (Beck, 2020). Accordingly, mediators were selected based on their theoretical relevance to core CBT processes targeted by the intervention (e.g., behavioural activation, problem-solving and cognitive restructuring) and evidence on mechanisms of change in low-intensity psychosocial interventions.

A mediator was defined as a variable on the causal pathway between the initial exposure to the intervention and the outcome (mean PHQ-9 score). It is therefore a variable that is causally influenced by the intervention and, in turn, causally influences the outcome (Lee et al., 2021). The selection of mediators was grounded in the theoretical framework underpinning the THP-TAP intervention and prior empirical evidence regarding key mechanisms of change in low-intensity psychosocial interventions (Table 1). Mediators were included in the final model if they were influenced by the intervention, influenced the outcomes or were associated with another mediator ($p < 0.05$). Using the above criteria for our two types of mediators, we can capture any potential mediator that is influenced by the intervention and influences either another mediator or the improvement in depressive symptoms.

Details of the mediators are given below:

*Number of sessions attended (M1)*: Participants who agreed to participate in the trial did not necessarily attend all sessions. We anticipated that the "dose" of the intervention would not only influence the extent to which symptoms of depression were

**Table 1.** Proposed mediators of THP-TAP

| Construct | Measure | Description |
|---|---|---|
| Perceived social support | Multidimensional Scale of Perceived Social Support (Zimet et al., 1988) | It comprises 12 items and 3 subscales related to social support from family, friends and significant others |
| Empathy and compassion | Empathy Scale for Lay Therapists (Liaquat et al., 2025) | The level of compassionate care delivered by peers to the intervention recipients will be assessed using a 12-item Empathy Scale for Lay Therapists. This scale has been developed by a panel of psychiatrists, psychologists and experts by experience |
| Patient activation | The PREMIUM Abbreviated Activation Scale (Manos et al., 2011; Singla et al., 2021) | It comprises 5 items and is based on the longer Behavioural Activation for Depression Scale and has been found to be reliable and valid for use in a similar population |
| Problem solving | Problem-solving confidence subscale of the Problem-solving Inventory (Heppner and Petersen, 1982) | It comprises 11 items and assesses perceived confidence for the use of problem-solving skills gained from a psychotherapeutic intervention |
| Cognitive restructuring | Cognitive restructuring subscale of the frequency of actions and thoughts scale (Terides et al., 2016) | The use of cognitive restructuring skills among the study participants will be assessed using the cognitive restructuring subscale of the frequency of actions and thoughts scale. This subscale comprises 3 items |

reduced, but also influence other mediators on the pathway to reduced symptoms of depression. To capture this effect, we created a mediator to reflect the number of sessions (M1: 0–8 sessions) attended in both the intervention and the control arms

*Patient activation (M2)*: Empowering individuals to take an active role in managing their mental health is central to behavioural activation approaches. The THP-TAP intervention was designed to enhance motivation and engagement through structured, peer-supported sessions. We used the PREMIUM Abbreviated Activation Scale, a five-item measure adapted from the longer Behavioural Activation for Depression Scale (Manos et al., 2011; Singla et al., 2021). This instrument has been validated in similar low-resource settings and captures the degree to which individuals feel capable of engaging in activities that support their well-being.

*Perceived social support (M3)*: Low levels of perceived social support have consistently been associated with increased vulnerability to perinatal depression (Beck, 2001; Dennis et al., 2004; Norhayati et al., 2015; Gelaye et al., 2016). In contexts of social adversity, such as those experienced by many women in Pakistan, stigma, poverty and limited family support may exacerbate feelings of isolation. The THP-TAP intervention encouraged peer-led engagement and emotional sharing, potentially enhancing perceived support. We assessed perceived social support using the culturally adapted version of the Multidimensional Scale of Perceived Social Support (Zimet et al., 1988). This 12-item scale evaluates support across three domains – family, friends and a significant other – with items rated on a seven-point Likert scale. Higher scores indicate stronger perceived support.

*Problem-solving confidence (M4a)*: Effective problem-solving is a core skill taught in the THP, enabling participants to break down overwhelming problems and generate actionable solutions. We assessed problem-solving confidence using the 11-item confidence subscale of the Problem-Solving Inventory (Heppner and Petersen, 1982). This subscale reflects participants' self-efficacy in applying learned strategies to real-life challenges, an important step in translating intervention content into behavioural change.

*Cognitive restructuring skills (M4b)*: Cognitive restructuring involves identifying and challenging unhelpful thoughts, a key mechanism in CBT-based interventions. The THP-TA intervention integrates these strategies in a culturally relevant and simplified format, supported through the digital app and peer facilitation. This construct was measured using the cognitive restructuring subscale of the Frequency of Actions and Thoughts Scale (Terides et al., 2016). This subscale includes three items evaluating the frequency with which participants engaged in reframing negative cognitions and adopting more adaptive thought patterns.

*Empathy and compassion from peer therapists (M5)*: A strong therapeutic relationship, often marked by empathy and compassion, is a known facilitator of psychological improvement, particularly in task-shared interventions. Peer therapists trained in the THP-TA programme were equipped with skills to convey empathy and foster trust with participants. To assess this dimension, we used the Empathy Scale for Lay Therapists, a 12-item instrument developed and validated through expert consensus, including psychiatrists, psychologists and service user representatives (Liaquat et al., 2025). The scale captures participants' perception of the emotional attunement and compassionate care provided by their peer therapist.

*Mediator-outcome confounders*: Due to the randomised nature of the exposure, it is not necessary to account for confounders of the association between exposure and outcome. However, for the current analyses, it was necessary to account for baseline confounders (unaffected by the intervention) that were associated with the mediator and/or the outcome. These mediator-outcome confounders can generate spurious correlations between the mediator and outcome when unadjusted for, potentially distorting these associations. We considered more than 30 potential baseline characteristics as potential confounders. Baseline values of all the mediators included in the models. Selection of our mediators was guided through DAGGITY, which help to remove colliders that can potentially bias our estimates (Shrier and Platt, 2008; Textor et al., 2011; Tennant et al., 2021).

### Statistical analysis

#### General

To better understand how the intervention influenced the mediators to improve symptoms of depression (Figure 1), we compare mediators and mediator outcome confounders with exposure to the intervention (control vs. intervention arm). To facilitate the comparison between mediators and mediator outcome confounders with recovery from depression, we dichotomised the outcome of mean PHQ-9 scores to recover/not recover from depression (PHQ-9 > 10 = not recovered).

#### Mediation analyses

We aimed to investigate the extent to which symptoms of depression measured at 6 months by the PHQ-9 questionnaire were explained by the above-listed mediators (Figure 1). To achieve this, we used an interventional (in)direct effects approach to mediation analysis that was used to understand population-level effects (Vansteelandt and Daniel, 2017). This approach provides an advanced causal inference framework that overcomes limitations of other mediation approaches, such as Structural Equation Modelling, that assume all mediators operate along distinct pathways or that all relationships are linear (Loh et al., 2020). Importantly, this approach also allowed us to test for moderated mediation, moderation by mediator-outcome confounders captured at baseline and exposure to the intervention on our outcome, as well as moderation by baseline variables and exposure to the intervention on our mediators. Findings are described following A Guideline for Reporting Mediation Analyses (AGReMA) guidelines for reporting mediation analyses (Lee et al., 2021).

#### Estimation and model fit

Our analyses were conducted using a series of regression models for mediators and outcomes that included non-linear terms and interactions between treatment allocation (intervention arm/control arm), mediator-outcome confounders and mediators. Estimation of Interventional Indirect Effects was based on Monte Carlo integration using a 1,000-fold expanded dataset (Vansteelandt and Daniel, 2017). The expanded dataset was created in five steps (Supplementary Material). The bootstrap also accounted for clustering at the primary health clinics. Bias-corrected confidence intervals (CIs) were based on non-parametric bootstrapping with 1,000 resamples (Vansteelandt and Daniel, 2017).

Briefly, we decomposed the total effect of the THP-TAP intervention into interventional indirect effects via each of the posited mediators as well as via the direct effect (e.g., no mediation). Our regression models simultaneously included all mediators, mediator-outcome confounders and relevant interactions and non-linearities. The indirect effect via a particular mediator can be interpreted as the average change in the potential outcome (symptom of depression at 6 months) resulting from shifting the

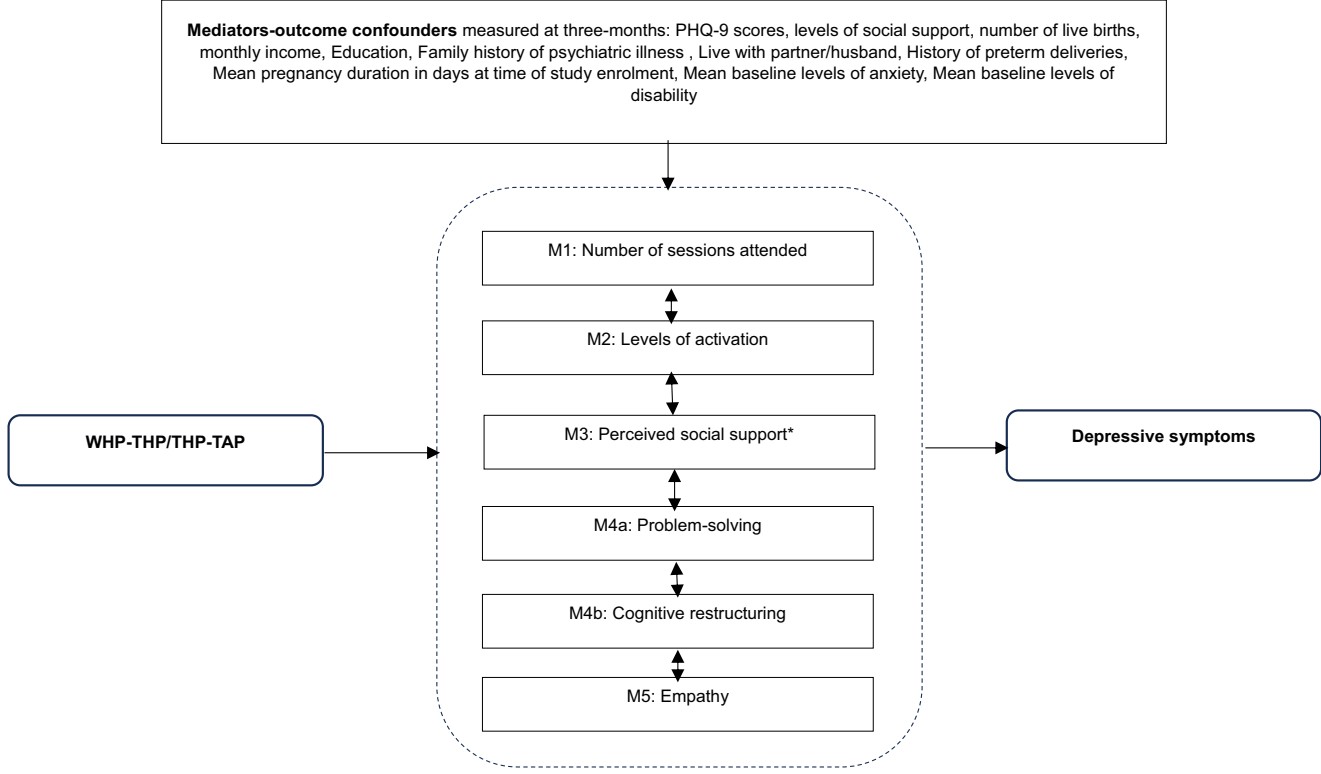

**Figure 1.** Causal mediation model demonstrating the proposed mediating pathways through which the ENHANCE intervention may improve symptoms of depression.
*M3: Perceived social support yielded statistically significant indirect mediaton effects: B = –0.072 (95% CI: –0.170, –0.018).

counterfactual distribution of that mediator from the exposed status (e.g., levels of social support in the intervention arm) to the unexposed status (levels of social support in the control arm), while setting each of the remaining mediators to random draws from either the exposed or unexposed group, depending on the specific decomposition.

### Assumptions

Interventional Indirect Effects have important underlying assumptions that influence the validity of findings if violated: consistency, no interference and no unmeasured confounders (Vansteelandt and Daniel, 2017). Due to the randomised nature of the study, assumptions of no unmeasured confounders between treatment arm and each mediator, and between treatment arm and outcome, are fulfilled. However, there is a possibility of unmeasured mediator-outcome confounders. To address this, we constructed a Directed Acyclic Graph identifying all potential measured and unmeasured confounders. As the trial captured multiple potential confounders, we anticipated that the chance of unmeasured mediator-outcome confounders was small. The "no interference" assumption infers that one participant's outcome does not influence another participant's outcome – for example, in a group-psychological intervention for perinatal depression, a large reduction in symptoms for one woman may influence other women to also experience symptom reduction. Given that the parent trial was a clustered randomised trial, we assumed that contact between participants was minimal. The consistency assumption states that an individual's potential outcome under their observed exposure history is precisely their observed outcome (Hernán and VanderWeele, 2011; VanderWeele and Hernán, 2013). Consistency is usually guaranteed in randomised experiments because the application of exposure to the participant is under the control of the investigators (Cole and

Frangakis, 2009). Given that the intervention was delivered via a digital tool will help to ensure the consistency assumption was upheld.

### Missing data
At baseline, 980 women were recruited to the study. However, there were only 823 (84.3%) women with complete data for this analysis. Due to the non-random nature of missing data (e.g., stillbirth and neonatal death), this mediation analysis includes complete data only.

Please see Supplementary File for further details on the methods.

## Results

### General

Please refer to Supplementary Table 1 for a detailed overview of participants' baseline characteristics and to Supplementary Figure 1 for the CONSORT flow diagram.

Table 2 below demonstrates that the mediators and mediator-outcome confounders were generally balanced between treatment arms. Perceived levels of social support appeared to be greater at baseline in the WHO-THP arm, shifting to be greater at 3 months in the THP-TAP arm. Women in the WHO-THP arm also had slightly higher levels of education, compared to women in the THP-TAP. Lastly, women in the THP-TAP arm had a higher mean income, compared to women in the WHO-THP arm.

Table 3 below demonstrates differences between some mediators and mediator-outcome confounders in recovery from depression (PHQ-9 < 10) at the 6-month follow-up visit. Levels of activation and avoidance, perceived social support, cognitive restructuring and empathy from delivery agent were higher in women who recovered

**Table 2.** Unadjusted comparison between mediators and mediator-outcome confounders and allocation to the control and intervention arm using complete data

| | Control arm (n = 408) | Intervention arm (n = 418) |
|---|---|---|
| **Mediators captured at 3 months** | | |
| M1: Mean number of sessions attended (SD) | 7.9 (0.4) | 7.5 (1.4) |
| M2: Mean levels of activation and Avoidance (SD)[a] | 14.5 (3.5) | 14.3 (3.1) |
| M3: Mean levels of perceived levels of social support (SD)[b] | 46.1 (12.2) | 47.5 (10.6) |
| M4a: Mean levels of problem-solving skills (SD)[c] | 13.9 (3.4) | 13.7 (3.7) |
| M4b: Mean levels of cognitive solving skills (SD)[d] | 8.4 (3.4) | 8.3 (3.3) |
| M5: Mean levels of empathy (SD)[e] | 31.1 (5.7) | 31.5 (5.2) |
| **Mediator outcome confounders captured at 3 months** | | |
| Mean baseline PHQ-9 scores (SD)[f] | 16.2 (4.5) | 16.9 (4.6) |
| Mean baseline levels of perceived social support (SD)[b] | 35.1 (13.9) | 33.2 (13.9) |
| Mean parity (SD) | 2.6 (1.5) | 2.6 (1.5) |
| Median monthly income (interquartile range) | 25,000 (15,000, 35,000) | 25,000 (15,000, 35,000) |
| Education<br>No education<br>Primary<br>Middle school<br>Matric<br>Intermediate school<br>University | 34 (8.3)<br>32 (7.8)<br>84 (20.6)<br>151 (37.0)<br>66 (16.2)<br>41 (10.1) | 39 (9.3)<br>51 (12.2)<br>87 (20.8)<br>130 (31.1)<br>61 (14.6)<br>50 (12.0) |
| Family history of psychiatric illness | 49 (12.0) | 49 (11.7) |
| Live with partner/husband | 371 (90.9) | 381 (91.4) |
| History of preterm deliveries | 17 (4.2) | 15 (3.6) |
| Mean pregnancy duration in days at time of study enrolment (SD) | 154 (35.6) | 148 (36.7) |
| Mean baseline levels of anxiety (SD)[g] | 11.8 (4.1) | 11.7 (3.7) |
| Mean baseline levels of disability (SD)[h] | 19.8 (8.5) | 20.6 (8.7) |

[a]The PREMIUM Abbreviated Activation Scale. All five items are assessed on a scale of 0 ("*not at all*") to 5 ("*yes, completely*") for a total continuous score of 25. Higher scores indicated increased levels of activation and avoidance (Rahman et al., 2023).
[b]Multidimensional Scale of Perceived Social support (MSPSS). The MSPSS scores range from 12 to 84. Higher scores indicating increased levels of social support (Zimet et al., 1988).
[c]Problem-solving confidence subscale of the Problem-solving Inventory (7-items, score range 7–28). Higher scores indicate higher levels of problem solving skills (Heppner and Petersen, 1982).
[d]Cognitive restructure subscale of the Frequency of Actions and Thoughts Scale. Scores range from 0 to 12. Higher scores indicate higher levels of cognition (Terides et al., 2016).
[e]Empathy Scale for Lay Therapists 12-items. Score range from 0 to 36. Higher scores indicate higher levels of empathy (Liaquat et al., 2025).
[f]Patient Health Questionnaire 9-items (PHQ-9). Each of 9 items scored 0–3 → total range = 0–27. Higher scores indicate more severe forms of depression (Kroenke et al., 2001).
[g]Generalised Anxiety Disorder 7-Items (GAD-7). Total range = 0–21. Higher scores indicated increased levels of anxiety (Spitzer et al., 2006).
[h]WHO Disability Assessment Schedule 2.0 (WHO-DAS). Scores ranged from 0 to 52. Higher scores indicate increased disability (Üstün et al., 2010).

from depression. At baseline, women with higher levels of education, monthly family income and cohabiting with a partner/husband were more likely to recover from depression. At baseline, women with a family history of a psychiatric disorder, history of preterm deliveries, higher mean duration of pregnancy, levels of symptoms of depression and anxiety and levels of disability were less likely to recover from depression.

### *Mediation analyses*

There was some evidence of moderated mediation whereby women with more severe symptoms of perinatal depression at baseline and lower levels of perceived social support (M3) at the 3-month follow-up had more severe symptoms of depression at the 6-month follow-up (p = 0.050). Similar moderated mediation was also suggested for empathy (M5). Specifically, women with more severe symptoms of depression at baseline and lower levels of empathy (M5) at the 3-month follow-up had more severe symptoms of perinatal

depression at the 6-month follow-up compared to women with fewer symptoms of depression (p = 0.05).

Table 4 describes the total causal effect, and the contribution of each of the indirect effects and the direct effect to the total causal effect. At 6 months, of the total mean difference in PHQ-9 scores between the intervention and control arm (adjusted mean difference in PHQ-9 scores: −0.214, bias-corrected 95% CI: −0.835, 0.437), 33% was mediated through perceived social support (M3 [adjusted mean difference in PHQ-9 scores attributable to perceived social support: −0.072, −0.170, −0.018], Table 4). In practice, this suggests that women receiving the digital intervention, THP-TAP, delivered by a peer had increased levels of social support compared to women who received the WHO-THP intervention by the LDWs, which mediated an improvement in symptoms of depression. There was no evidence that the ENHANCE intervention improved symptoms of depression through any of our other mediators. This suggests that the THP-TAP and the WHO-THP were similar with respect to their effects on the potential mediators.

**Table 3.** Unadjusted comparison between mediators and mediator-outcome confounders with recovery from depression (PHQ-9 < 10) at the 6-month follow-up using complete data in the intervention arm only

| | No (*n* = 68, 9.7%) | Yes (758, 90.3%) |
|---|---|---|
| Mediators captured at 3 months | | |
| Number of sessions attended (M1) | 7.7 (1.4) | 7.7 (1.1) |
| Levels of activation and avoidance (M2)[a] | 12.8 (3.4) | 14.5 (3.3) |
| Perceived levels of social support (M3)[b] | 40.6 (14.1) | 47.4 (11.0) |
| M4a: Problem-solving skills (M4a)[c] | 15.9 (3.7) | 13.6 (3.7) |
| M4b: Cognitive-solving skills (M4b)[d] | 6.4 (3.6) | 8.6 (3.3) |
| M5: Empathy (M5)[e] | 28.8 (5.9) | 31.6 (5.8) |
| Mediator outcome confounders captured at 3 months | | |
| Mean baseline PHQ-9 scores (SD)[f] | 18.1 (4.5) | 16.4 (4.5) |
| Mean baseline levels of social support (SD)[b] | 33.6 (13.7) | 34.2 (13.9) |
| Mean number of live births (SD) | 2.9 (1.8) | 2.6 (1.4) |
| Median monthly income (interquartile range) | 20,000 (15,000, 30,000) | 25,000 (16,000, 35,000) |
| Education<br>  No education<br>  Primary<br>  Middle school<br>  Matric<br>  Intermediate school<br>  University | <br>9 (13.2<br>11 (16.2)<br>21 (30.9)<br>18 (26.5)<br>5 (7.4)<br>4 (5.9) | <br>64 (8.4)<br>72 (9.5)<br>150 (20.0)<br>263 (34.7)<br>122 (16.1)<br>87 (11.5) |
| Family history of psychiatric illness | 14 (20.6) | 84 (11.1) |
| Live with partner/husband | 68 (85.0) | 685 (91.8) |
| History of preterm deliveries | 4 (5.9) | 28 (3.7) |
| Mean pregnancy duration in days at time of study enrolment (SD) | 154.7 (36.4) | 151.1 (36.2) |
| Mean baseline levels of anxiety (SD)[g] | 13.5 (4.2) | 11.6 (3.9) |
| Mean baseline levels of disability (SD)[h] | 22.2 (9.0) | 20.0 (8.5) |

[a]The PREMIUM Abbreviated Activation Scale. Higher scores indicated increased levels of activation and avoidance.
[b]Multidimensional Scale of Perceived Social support (MSPSS). Higher scores indicating increased levels of social support (Zimet et al., 1988).
[c]Problem-solving confidence subscale of the Problem-solving Inventory. Higher scores indicate higher levels of problem solving skills (Heppner and Petersen, 1982).
[d]Cognitive restructure subscale of the Frequency of Actions and Thoughts Scale. Higher scores indicate higher levels of cognition (Terides et al., 2016).
[e]Empathy scale for Lay Therapists 12-items. Higher scores indicate higher levels of empathy (Liaquat et al., 2025).
[f]Patient Health Questionnaire 9-items (PHQ-9). Higher scores indicate more severe forms of depression (Kroenke et al., 2001).
[g]Generalised Anxiety Disorder 7-Items (GAD-7). Higher scores indicated increased levels of anxiety (Spitzer et al., 2006).
[h]WHO Disability Assessment Schedule 2.0 (WHO-DAS). Higher scores indicate increased disability (Ustün et al., 2010).

**Table 4.** Total causal effect and interventional in(direct) effects for the ENHANCE programme at 6 months

| Effects | Estimates and bias-corrected 95% CI[a,b,c,d] |
|---|---|
| Total effect of THP on improved symptoms of perinatal depression | −0.214 (−0.835, 0.437) |
| Direct effect | −0.092 (−0.616, 0.473) |
| Indirect effect through session attendance (M1) | −0.072 (−0.205, 0.026) |
| Indirect through levels activation and avoidance (M2) | 0.017 (−0.045, 0.074) |
| Indirect effect improved levels of perceived social support (M3) | −0.072 (−0.170, −0.018) |
| Indirect effect through problem solving and cognitive restructuring (M4) | 0.004 (−0.108, 0.010) |
| Indirect through empathy (M5) | −0.017 (−0.094, 0.007) |
| Indirect effect through the mutual dependence of mediators on one another | −0.010 (−0.040, 0.003) |

[a]Estimates have been adjusted for mediator-outcome confounders captured at baseline including: PHQ-9 scores, levels of perceived social support, parity, income, education, living in a nuclear family, having a family history of mental illness, cohabiting with partner/husband, have a previous preterm birth, number of people living in the household, age, levels of anxiety and levels of disability.
[b]Interactions between mediator-outcome confounders includes perceived levels of social support (M3) and baseline PHQ-9 scores, empathy (M5) and baseline PHQ-9 scores.
[c]Estimation for the different effects was based on Monte Carlo integration using 1,000-fold expanded dataset.
[d]Bias-corrected confidence intervals were based on nonparametric bootstrap with 1,000 resamples adjusting for clustering.

## Discussion

This study reports mediation analysis to determine whether THP-TAP shares the same CBT-specific and common-factor mechanisms as the standard established WHO-THP, or whether it achieves its effects via different pathways. Three main findings emerge. First, depressive symptoms at 6 months were statistically indistinguishable between arms, confirming the non-inferiority of the digital technology-assisted peer-delivered innovation for sustained mood outcomes. Second, exposure to the digital THP-TAP led to improvements in symptoms of depression mediated by perceived social support, compared to exposure to the WHP-THP. This underscores the technology-assisted format's enhanced capacity to mobilise social networks in recovery. Third, there was no difference in symptom reduction for all other mediators, including the number of sessions attended, empathy and putative CBT-specific mediators (behavioural activation, problem-solving and cognitive restructuring).

### *Interpretation in relation to previous work*

These mediation findings provide nuanced insights beyond our previous cRCT (Rahman et al., 2025), revealing that the THP-TAP achieves non-inferiority through mechanisms that differ subtly from those of the WHO-THP model. Whereas the standard THP relies on LHWs, THP-TAP employs peers embedded in the same villages. This proximity plausibly enhances credibility, trust and the peers' capacity to mobilise husbands, mothers-in-law and neighbours to support behavioural activation homework and problem solving (Atif et al., 2017; Rahman et al., 2021; Rahman et al., 2023). The modest but significant perceived social support pathway is therefore theoretically coherent and concordant with prior qualitative work in Pakistan and India, where women emphasised "feeling understood" and "practical help from family" as critical to recovery (3, 6). These results align with earlier mediation analyses of the WHO-TAP and its peer-delivered adaptation Thinking Healthy Programme-Peer delivered (THPP), where patient activation and social support, but not mother–child attachment, were identified as key mediators of treatment effect in trials conducted in India and Pakistan (Singla et al., 2021; Elias et al., 2024).

Our findings are also consistent with the growing evidence base supporting the delivery of psychotherapeutic interventions through digital and remote modalities for perinatal populations. For instance, the Scaling Up Maternal Mental healthcare by Increasing access to Treatment (SUMMIT) trial demonstrated that telemedicine-delivered behavioural activation was non-inferior to in-person psychotherapy for perinatal depression and anxiety across a range of outcomes, including depressive symptom reduction, therapeutic alliance and patient satisfaction (Singla et al., 2025). These results highlight that core relational mechanisms of therapy, such as empathy, alliance and perceived social support, can be effectively preserved within virtual care models. Importantly, SUMMIT also found comparable outcomes between specialist and non-specialist providers.

Our study builds upon and extends these insights to a different delivery model and implementation context. Whereas SUMMIT examined synchronous, therapist-led psychotherapy via telemedicine in high-income settings, our intervention evaluated a technology-assisted, peer-delivered model implemented in a low-resource community context. In such settings, digital tools not only serve as a conduit for care but also as a fidelity mechanism that standardises delivery, guides peer providers step-by-step and embeds therapeutic components that might otherwise depend on specialist expertise.

The focus on empathic engagement in our study provides an important complement to traditional measures of therapeutic alliance: It captures the affective and attitudinal aspects of relational quality that are central to effective psychosocial care, even when sessions are guided by structured digital content rather than conventional therapist–client interactions. Taken together, our results suggest that the relational foundations of psychotherapy – typically thought to rely on in-person human contact – can be maintained in digitally mediated, task-shared formats. This convergence with SUMMIT reinforces the notion that digital delivery does not necessarily compromise therapeutic quality. Rather, when supported by structured training, supervision and evidence-based digital tools, non-specialists and peers can achieve meaningful engagement and therapeutic outcomes that parallel those of specialist- or telemedicine-delivered interventions.

The absence of detectable mediation via CBT skill acquisition warrants reflection. One interpretation is measurement insensitivity: Session-level adherence checklists may capture the delivery of CBT elements but not the depth of internalisation by participants. Alternatively, THP-TAP's animated storylines may have standardised CBT delivery so effectively that between-arm variance in skill acquisition (including cognitive restructuring, problem solving and behavioural activation) was too small to mediate outcome differences. Under either scenario, the data suggest that fidelity to specific CBT techniques was successfully maintained across platforms, supporting the premise that automation can shield core content from "programme drift" even when supervision intensity is reduced (4).

### *Mechanistic and practical implications*

Taken together, the findings imply a division of labour between technology and human agents. The app guarantees consistent CBT dosage and sequencing, while peers with lived experience provide empathic resonance and activate existing social networks. This distinction is particularly important: Social support was perceived by perinatal women as less accessible in the LHW-delivered WHO-THP, where competing public health responsibilities constrain opportunities for relationship building. For health systems grappling with LHW overload, the technology-assisted peer-delivered intervention offers two advantages: (i) It preserves therapeutic potency without further taxing professional cadres, and (ii) it yields incremental gains in social connectedness, a protective factor with benefits beyond depression per se. From an implementation standpoint, the modest magnitude of the social-support pathway should explicitly cultivate community-engagement skills, and digital content could be updated to include structured prompts that encourage involvement of key relatives. Because cost-effectiveness models already indicate lower per-patient costs for THP-TAP once equipment is amortised (6), even small additional gains in social functioning are likely to improve value for money.

### *Strengths and limitations*

This mediation study possesses several methodological strengths that enhance the credibility and relevance of its findings. First, it is embedded within a large, single-blind, pragmatic cluster-randomised trial spanning 70 villages. By utilising existing LHW infrastructure and community-nominated peers, the research is conducted in real-world health system conditions, which contributes to its external validity. Moreover, the use of robust causal mediation analysis and exploration of mediator interdependencies improves the robustness of results. Finally, linguistic validation of mediation instruments,

as well as co-production with service users and engagement of district health authorities, enhanced the psychometric soundness of interviews.

However, this study also has several limitations. Although cluster randomisation reduced contamination, social ties across neighbouring villages may still have diluted between-arm contrasts. Outcomes and mediators were captured by self-report – inviting shared-method and social-desirability bias – and the final mediator assessment coincided with the primary outcome, so temporal ordering is inferred rather than observed. The parent trial was powered for non-inferiority, and not mediation, limiting sensitivity to detect small indirect effects. Additionally, findings may not generalise to urban LMIC settings with different workforce and digital infrastructure profiles. Another limitation is that the assessment of key behavioural outcomes relied solely on self-reported activation levels rather than incorporating objective measures such as those obtainable through wearable devices.

### Future research

The findings endorse technology-assisted, peer-delivered THP-TAP as a service-ready alternative to the face-to-face LHW model. Because the App standardises CBT content and the peers are drawn from the same communities, districts can redeploy scarce supervisory resources towards recruitment, brief tablet-based training and light-touch quality monitoring rather than continuous skills coaching. The additional benefit in perceived social support suggests that embedding lived experience peers not only preserves therapeutic potency but also addresses a key social risk factor in the perinatal period. Ministries of Health and Non-governmental organizations (NGOs) implementing maternal-child-health packages could therefore integrate THP-TAP into routine antenatal and postnatal visits, use peer facilitators to engage husbands and mothers-in-law and leverage existing tablet roll-outs for immunisation or community surveillance to keep marginal costs low.

For researchers, the modest but significant social support pathway invites a sharper focus on relational mechanisms in low-intensity CBT. Future trials should incorporate objective and ecological measures of network engagement, experimentally manipulate "dosages" of digital content versus peer contact and test whether augmenting social support modules yields incremental benefit. Longitudinal follow-ups beyond the first post-partum year are needed to examine whether early gains translate into sustained maternal functioning and improved child developmental trajectories. Finally, adaptive platform trials that embed mechanism-driven optimisation cycles could refine both digital assets and peer-training curricula, moving the field towards precision task-sharing in global mental health care.

### Conclusions

Digital THP delivered by peers with lived experience is clinically equivalent to, and on some social dimensions superior to, standard WHO-THP delivered by trained community health workers (CHWs). Mediation analysis indicates that increased perceived social support, rather than differential CBT skill mastery, constitutes an effective pathway of additional benefit. These findings bolster the case for integrating technology-assisted, peer-delivered psychosocial interventions into overstretched maternal health services and highlight community-embedded support as a modifiable target for future optimisation.

Furthermore, these findings underscore that technology-enabled interventions should be designed to strengthen, rather than replace, human connection. In this study, the digital platform functioned as a facilitator of supportive peer–mother relationships, highlighting the importance of maintaining interpersonal processes at the core of scalable psychosocial care.

**Open peer review.** To view the open peer review materials for this article, please visit http://doi.org/10.1017/gmh.2025.10109.

**Supplementary material.** The supplementary material for this article can be found at http://doi.org/10.1017/gmh.2025.10109.

**Author contribution.** AW, AR and SS designed the study. NA and AR designed the intervention. NS conducted the data analysis. NS and AW interpreted the study results and wrote the initial draft of the manuscript. AM and HN co-developed the intervention and oversaw the field work. DW and AN provided statistical oversight. All authors critically reviewed the manuscript and approved the final manuscript for submission.

**Competing interests.** The authors declare none.

**Ethical statement.** The authors assert that all procedures contributing to this work comply with the ethical standards of the relevant national and institutional committees on human experimentation and with the Helsinki Declaration of 1975, as revised in 2013. Ethical approval was secured from the Research Ethics Committees of the Human Development Research Foundation (Pakistan), the University of Liverpool (ethical approval: UoL001668) and the National Bioethics Committee (Pakistan).

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
