## [Reviewer Report]

Thank you for the opportunity to review this manuscript. This is an important contribution to the growing literature on mediation analysis in treatment trials. My general comments are below.

INTRODUCTION

• It is unclear how the sentences about voltage drop and programme drift relate to this manuscript on mediation analysis. Please remove and clarify how this relates to the primary aim

• Did the authors use a theoretical framework to determine their proposed mediators of analysis

• More information is required about the ENHANCE non-inferiority trial, e.g., please clarify on how mediators can be examined in trials where non-inferiority results were found.

METHODS

• The authors mentioned that they use the PREMIUM Abbreviated Activation Scale to assess patient activation and refer to their Trials protocol. It would be more appropriate to refer to the original source.

• Given the larger ENHANCED trial was a non-inferiority trial, and causal mediation typically requires differential effects of the treatment on the proposal mediator (see Vanderweele, 2016), how did the authors go about conducting this analysis in the proposed study? Further discussion is required in both the methods and results section

• It is unclear whether the authors estimated all potential mediators simultaneously and if so, what model (e.g., structured equation modelling) was used

• Please add the estimated contribution of each potential mediator of the total effect

• Please clarify whether the authors measured and were able to control for proposed mediators at baseline.

RESULTS

• Related to my previous comment re: non-inferiority, the results are difficult to interpret. It is unclear whether differential effects were found for the proposed mediator of social support. E.g., the authors write “Table 3 demonstrates that at six months, there was no difference in mean PHQ-9 scores between participants receiving the THP-TA and participants receiving the WHO-THP (total causal effect: adjusted mean difference in PHQ-9 scores: -0.214, bias-corrected 95% CI: -0.835, 0.437). There was some evidence to support mediation through perceived social support (M3) (adjusted mean difference in PHQ-9 scores attributable to perceived social support: -0.072, -0.170, -0.018) (Table 4). In practice, this suggests that women receiving the digital intervention delivered by the a peer who has experience as a mother, had increased levels of social support compared to women that received the original THP intervention by the closely supervised Lady Health Workers, that mediated an improvement in symptoms of depression.”

• Please report the estimated contribution of each potential mediator of the total effect

DISCUSSION

• The authors mention that “the mediational findings…revealing that the THP-TAP achieves equivalent outcomes”; however, this was a non-inferiority trial not an equivance trial. Therefore, it is not appropriate to report that ‘equivalent outcomes’ were achieved.

• The discussion is missing a full interpretation of the results in relation to other studies. For example:

1. The authors include a section entitled ‘interpretation in relation to previous work’; however, (1) key references of other mediation analyses in perinatal populations are missing that have been previously published in this journal examining mediators and moderators for perinatal mental health in LMICs (Elias, Seward, Lund, 2024, Cambridge Prisms: Global Mental Health) and (2) from this group who demonstrated that activation and perceived support were key mediators in treatment effects in reducing perinatal depressive in a similar region (e.g., Singla et al., British Journal of Psychiatry). This is important to fully interpret the current findings

2. it is not clear how these findings related to other digitally-delivered psychotherapeutic interventions for perinatal populations and otherwise. While the authors should be commended for their measurement of empathy among providers, typically therapeutic alliance are assessed in psychological treatment trials (e.g., the SUMMIT Trial compared telemedicine to in-person psychotherapy for perinatal populations with depressive and anxiety symptoms and found non-inferiority across a number of outcomes including therapeutic alliance and patient satisfaction).

3. Can the authors elaborate why only perceived support was found as a mediator and not other key elements such as cognitive restructuring

• These additions would improve the manuscript substantially to provide the reader with a full interpretation of the current findings in relation to the extant literature.

• Another limitation is that the findings on key behavioral measures are limited to self-report activation levels vs. objective measures using wearables.

---

## [Reviewer Report]

Please remove abbreviations from the abstract

Introduction- well written

Methodology: A detailed flowchart illustrating recruitment, retention, and attrition at different time points would be beneficial for readers.

Socio-demographic characteristics of the participants, apart from the variables that were considered as mediators, were not described in the article.

Results and discussion were good

Overall, the article seems fine! Minor revisions are recommended.

---

## [Editor Report]

Dear Authors,

Please see the reviewers' comments and suggestions. I have read them and found them to be highly relevant to your study, which will enhance the scope of the findings. Therefore, I encourage you to consider responding to the comments and resubmitting the manuscript.

Thanks and regards,

Thomas

---

## [Reviewer Report]

Thank you to the co-authors for addressing most of my comments, which are largely addressed and this manuscript is much improved.

A few additional (minor) comments :

1) suggest that the authors add a theoretical framework, whether it be social learning theory or another health behaviour change model. The current one referred to in the responses is not exactly a theoretical framework but rather a rationale for why these mediators were selected.

2) Table 1 should refer to “Proposed” Mediators. Also, citations/references for specific scales could be added for completeness.

3) Figure 1 is important but it is unclear which variables in this multiple mediation model were found to be mediators of the intervention. This could be clarified by combining the results from one of the tables and Figure 1. Please also review spelling throughout (an “N” is missing in the “ENHANCE” in the title)

4) A thorough review of the manuscript may be warranted. There are comments in the revised draft (e.g., ‘This could be vanderweele, 2016’) that seem to be for the co-author group, not the reviewer group?

5) the connection between ENHANCE the THP is not always clear and various labels seem to be used to describe the intervention arm e.g., ENHANCE, THP-TAP, etc. Suggest that this is clearly defined in the abstract and early in the manuscript and only one term is used to label the intervention arm thereafter. Relatedly, the current definition of ENHANCE in the abstract is vague i.e., “technology-assisted digital adaptation of the Thinking Healthy Programme”. What does this mean? is it simply that the intervention was delivered virtually? that conversational agents were used to deliver the intervention? that this was a guided self-help app?

6) the number of sessions should be mentioned in the abstract if ‘number of sessions’

is a proposed mediator.

7) Table 2 requires possible ranges of scores for a given measure in the first column to give the reader a sense of what the mean scores mean for a given variable. Relatedly, suggest the authors clarify why they used the sum scores for the MSPSS vs. the average.

8) Discussion: How should we interpret the finding that precevied support was the sole mediator of treatment effects in this study (as found in most interventions for perinatal mental health globally) in light of evidence that technology use may be eroding human connection? I believe a comment on this point would be important in light of the study findings.

---

## [Reviewer Report]

The authors have adequately addressed the comments in the revised manuscript. Therefore, I don’t have any further comments.

---

## [Editor Report]

Dear Authors,

I appreciate your time and efforts in addressing the reviewer’s comments and suggestions. I am happy to share that the reviewers are overall satisfied with the revision, however, they also see scope for addressing some minor issues. I concur with them, and request you kindly consider addressing the issues and resubmitting the manuscript.

Sincerely,

Thomas

---

## [Editor Report]

Dear Authors,

On behalf of the reviewers and the editorial team, I appreciate you for revising the manuscript satisfactorily. I am happy to recommend the manuscript for acceptance and further processing.

I hope your association with Cambridge Prisms: Global Mental Health continues in the future.

With best wishes,

Thomas